# Cryopreservation of Indigenous Plums and Monitoring of Multiplication and Rooting Capacity of Shoots Obtained from Cryopreserved Specimens

**DOI:** 10.3390/plants12173108

**Published:** 2023-08-30

**Authors:** Tatjana Vujović, Tatjana Anđelić, Bojana Vasilijević, Darko Jevremović, Florent Engelmann

**Affiliations:** 1Fruit Research Institute, Kralja Petra I No. 9, 32000 Čačak, Serbia; tandjelic@institut-cacak.org (T.A.); bvasilijevic@institut-cacak.org (B.V.); djevremovic@institut-cacak.org (D.J.); 2Institute of Research for Development, 911 Av. Agropolis, P.O. Box 64501, CEDEX 5, 34394 Montpellier, France; florent.engelmann@ird.fr

**Keywords:** *Prunus domestica* L., plant vitrification solution, V cryo-plate, desiccation, D cryo-plate, regrowth, multiplication, rooting

## Abstract

The objective of this study is to assess the suitability of vitrification cryo-plate (V cryo-plate) and dehydration cryo-plate (D cryo-plate) methods for the long-term conservation of eight autochthonous *Prunus domestica* L. genotypes originating from the Balkan Peninsula region. In vitro shoot tips were briefly pre-cultured for 1 day at 23 °C in the dark on a medium containing 0.3 M sucrose and then embedded in calcium alginate gel within the wells of the aluminum cryo-plates. In the V cryo-plate protocol, dehydration was carried out at room temperature using the following vitrification solutions: original plant vitrification solution 2 (PVS2) and 90% PVS2 solution (for 20 and 40 min) and plant vitrification solution 3 (PVS3) (for 60 and 80 min). In the D cryo-plate protocol, desiccation was performed for 2, 2.5, or 3 h over silica gel at 23 °C. The effect of different treatments was evaluated by monitoring the regrowth of both non-frozen and cryo-preserved explants. After cryo-preservation, five genotypes achieved regrowth rates over 40% in at least one of the applied protocols, while two genotypes showed regrowth rates of around 10%. A significant improvement in regrowth success for all genotypes using both cryo-plate methods was achieved by pre-culturing shoot tips for 7 days on a medium containing 0.5 M sucrose in complete darkness at 4 °C. Shoots regenerated from cryo-preserved explants were further monitored in vitro. By the third subculture, they had not only regained but had even exceeded the multiplication capacity (index of multiplication, length of axial, and lateral shoots) of shoots regenerated from dissection controls. Following multiplication, the cryo-preserved shoots were successfully rooted and rooting ability was assessed by monitoring the percentage of rooting, number and length of roots, and height of rooted plantlets.

## 1. Introduction

European plums (*Prunus domestica* L.) include, among others, a large group of old Eastern European cultivars, primitive forms, and autochthonous biotypes (landraces) that are part of intangible heritage, tradition, customs, and legacy, as well as cultural identity. Botu et al. [1] consider *P. domestica* L. to be an indigenous species in the Balkans, given the large number of landraces that have been grown there for centuries. Most of these genotypes are grown on their own roots and are propagated both by suckers and by seeds; therefore, plum populations exhibit high heterogeneity [2]. High genetic variability is evident in different biological and productive traits, particularly in fruit characteristics (size, shape, color, texture, and aroma) and adaptability to different ecological conditions. Formerly, these multi-purpose plum genotypes were widely cultivated and used not only for fresh consumption but also for canning, drying, cooking, and processing into plum brandy [3]. However, this rich plum germplasm has been gradually replaced by improved newly-bred cultivars with superior pomological production and post-harvest features, which are consequently exposed to slight and irreversible genetic erosion and disappearance [4].

A high phenotypic diversity of old autochthonous plum cultivars and primitive cultivated landraces has also been described in the Balkan Peninsula region [2,5,6,7,8]. This diversity represents a good genetic basis for the selection of clones resistant to economically important diseases (Sharka, plum leaf blotch, and plum rust), as well as for different breeding programs for the development of new plum cultivars and *Prunus* rootstocks [5]. In addition, interest in growing old plum cultivars has increased due to their lower demands, higher adaptability to local agro-ecological conditions, and superior flavor and taste of their fruits, which make them suitable for low-input agriculture and organic production [9,10]. Therefore, considering the importance of this exceptionally rich plum germplasm, it is necessary to develop concepts for its collection, sustainable conservation, management, and utilization. However, like many other temperate fruit species, plums are genetically heterozygous and are propagated vegetatively. Because of their high heterozygosity, individuals with specific combinations of genetic traits, once identified and selected, cannot be regenerated through seeds [11]; instead, they are typically preserved in the field as active collections.

Other than collections maintained in the open field, contemporary plant biodiversity conservation programs involve employing ex situ strategies that enable the storage of biological materials in artificial environments (in vitro culture) and their reintroduction into natural habitats as required [12]. Apart from in vitro techniques intended for short- and medium-term conservation of vegetatively propagated plant species, rapidly developing cryo-preservation techniques that enable long-term storage of plant material in liquid nitrogen (LN) are currently considered integrated and complementary to classical plant conservation strategies, providing an additional guarantee against accidental loss of genetic resources [13]. Given the considerable heterozygosity in plum cultivars, clonal preservation, based on the cryo-preservation of shoot tips or axillary buds, is much more desirable than seed or pollen preservation [14,15].

In general, there are two possible methods for the cryo-preservation of plant tissues [16]. The two-step cooling method involves slow cooling of explants down to −40 °C, followed by fast cooling by immersion in LN. The second cryo-preservation method is based on vitrification of extra-cellular and intra-cellular plant tissue solutions during the ultra-fast cooling process. Both methods have relative advantages and disadvantages for long-term storage and the choice of the technique is dependent on the plant type, operator skills, expertise, available equipment, and facility [17].

Vitrification-based cryo-preservation techniques involve the treatment of explants with a moderately concentrated cryo-protectant solution, so-called loading solution (LS), and highly concentrated plant vitrification solutions (PVSs) to lower water content and enable ultra-fast cooling processes [18]. According to Niino and Aziraga [19], vitrification represents a successful freeze-avoidance mechanism for hydrated cells and tissues. Until now, some of the vitrification-based cryo-preservation techniques have been applied, with varying success, for the conservation of different *P. domestica* genotypes. Vitrification, one-step cooling, and encapsulation–vitrification techniques gave satisfactory results in cultivar ‘Regina Claudia’, while regrowth of explants conserved using the encapsulation–dehydration technique was significantly lower [20]. Shoot tips of cultivar ‘Torinel’, cryo-preserved by the droplet vitrification technique, also displayed high regrowth capacity [21]. However, autochthonous plum genotypes ‘Crvena Ranka’, especially ‘Sitnica’, had poor regrowth after retrieval from LN [22].

Vitrification methods using aluminum cryo-plates have the potential to become suitable protocols for recalcitrant plants and can facilitate the efficient implementation of cryo-storage and long-term maintenance of plant genetic resources in gene banks [23,24]. Dehydration of explants attached to cryo-plates in sodium alginate droplets is achieved using PVSs (V cryo-plate method) or air dehydration (D cryo-plate method). Investigations on the suitability of these two methods for cryo-preservation of *P. domestica* [25,26] confirmed that they could improve the regrowth of genotypes that displayed poor regrowth capacity after cryo-storage using other techniques. Therefore, this paper investigates the possibility of cryo-preserving in vitro-grown shoot tips of eight *P. domestica* genotypes using both V cryo-plate and D cryo-plate methods. Although originating from different locations in Serbia, the studied plums are of regional importance (Balkan Peninsula), as numerous cases of synonymy in traditional plum cultivars have previously been well documented [27]. Evaluation of the V and D cryo-plate methods was performed through the following steps: (i) comparison of different types of PVSs and treatment durations to find the most suitable treatment(s) that could achieve the correct balance between toxicity and adequate dehydration of samples in the protocol for the V cryo-plate technique, (ii) optimization of the desiccation (air dehydration) step by modifying treatment duration in the D cryo-plate technique, (iii) evaluation of the effect of prolonged pre-culture of shoot tips on a medium enriched with sucrose on cryo-preservation success; (iv) monitoring of regrowth of both control (non-frozen) and cryo-preserved explants after retrieving from LN, and (v) monitoring of multiplication and rooting ability of shoots regenerated from cryo-preserved explants and their comparison to adequate controls.

## 2. Results

### 2.1. Monitoring of Growth Recovery

Regrowth percentages of both control (non-frozen; −LN) and cryo-preserved (+LN) explants, monitored at the end of the 6th week, are presented in Table 1 and Table 2 for the first and second setups of the cryo-preservation experiments, respectively.

#### 2.1.1. The First Experimental Setup

At the end of the 6-week monitoring period in the first set of experiments, the regrowth percentages of dissection, pre-growth, and loading controls reached 100% in most of the genotypes (Table 1). The only exceptions were dissection and loading controls in ‘Trnovača’ (90% regrowth for both), C4 loading controls in ‘Dragačevka’ and ‘Cerovački Piskavac’ (80% and 90%, respectively), and LS1 loading controls in ‘Crnošljiva’, ‘Dragačevka’, and ‘Trnovača’ (50%, 60%, and 50%, respectively). However, regrowth of desiccation and dehydration controls was higher (significantly higher in most of the treatments) than regrowth of corresponding cryo-preserved explants. 

In the V cryo-plate protocol, plums exhibited low regrowth potential after dehydration with PVS2 solution followed by LN exposure, with regrowth lower than 20% in most genotypes (Table 1). Regeneration percentages between 20% and 30% were only noted in cryo-preserved explants of ‘Belošljiva’, ‘Sitnica’, ‘Požegača’, ‘Moravka’, and ‘Cerovački Piskavac’ (Figure 1f) when dehydrated for 40 min with this VS. Regarding 90% PVS2 (PVS A3) treatments, cryo-preserved explants of ‘Moravka’ displayed significantly higher regrowth potential, but only for a 40 min treatment (62.5%; Figure 1d). Regrowth higher than 30% was observed only in ‘Belošljiva’ (37.5%; Figure 1a) and ‘Požegača’ (33.3%; Figure 1c) for cryo-preserved explants also treated for 40 min with PVS A3 and in ‘Crnošljiva’ (30.0%) for those treated for 20 min with the same VS. All other genotypes displayed very low regeneration (lower than 20%) after cryo-preservation. Treatment of explants with PVS3 solution resulted in increased regrowth in cryo-preserved explants of ‘Sitnica’ (45.8% for 80 min treatment; Figure 1b), as well as in genotype ‘Crnošljiva’ (42.9% for 80 min treatment; Figure 1e). Genotype ‘Moravka’ also displayed satisfactory regeneration (48.1%) after dehydration with PVS3 for 60 min. Regrowth of cryo-preserved explants of ‘Belošljiva’ and ‘Cerovački Piskavac’ was between 20% and 30%, while other genotypes exhibited regeneration lower than 20% (Figure 1g,h).

Considering regrowth success after cryo-preservation using the D cryo-plate method (Table 1), plum genotypes could be divided into three main groups: (i) genotypes that displayed acceptable regrowth (‘Belošljiva’, 29.2–41.7%; ‘Sitnica’, 25.0–41.6%; ‘Požegača’, 40.0–65.0%) following LN treatment, the highest regrowth percentages achieved with the shortest treatment duration (Figure 2a–c); (ii) genotypes that displayed low (under 30%) regrowth after cryo-preservation (‘Crnošljiva’, 12.5–16.7%; ‘Moravka’, 10.0–29.2%; ‘Cerovački Piskavac’, 0.0–25.0%), the highest regrowth percentages achieved with the longest treatment duration (Figure 2d–f); (iii) genotypes in which it was not possible to regenerate shoots after cryo-preservation (‘Dragačevka’ and ‘Trnovača’).

The cryo-preservation method used, as well as the duration of desiccation or dehydration and the type of VS used, did not influence the growth characteristics of shoot tips regenerated from cryo-preserved explants of autochthonous plums. The first signs of shoot regeneration were observed 3 weeks after explant transfer to regrowth medium. By the sixth week, the regrown plantlets, although with short stems, were normally developed with no signs of hyperhydricity (Figure 1 and Figure 2). 

#### 2.1.2. The Second Experimental Setup

The regrowth capacity of both dehydration/desiccation controls and corresponding cryo-preserved shoot tips was improved in all examined plum genotypes by prolonged pre-culture of shoot tips for 7 days on a medium enriched with 0.5 M sucrose alone (V cryo-plate protocol) or in combination with a change in the loading solution applied (D cryo-plate protocol). As for the dehydration/desiccation controls, this increase was particularly notable in treatments that had previously shown regrowth rates lower than 50%, such as: (i) 80 min PVS3 treatment in ‘Cerovački Piskavac’ (from 30% to 80%) and ‘Trnovača’ (from 11.1% to 70%); (ii) 2 h desiccation in ‘Dragačevka’ (from 10% to 76.7%) and ‘Trnovača’ (from 0% to 63.3%); (iii) 3 h desiccation in ‘Moravka’ (from 50% to 90%), ‘Crnošljiva’ (from 20% to 63.3%), and ‘Cerovački Piskavac’ (from 30% to 86.7%).

The regrowth capacity of cryo-preserved explants increased in line with the increase in the ability of non-frozen shoot tips to regenerate. However, the regrowth rates of cryo-preserved explants remained significantly lower than those observed in corresponding dehydration or desiccation controls. The only exception was observed in ‘Sitnica’ after 2 h of desiccation, where the regrowth rate was 83.3% in controls and 75% in cryo-preserved explants. 

Comparing the regeneration percentages between cryo-preserved explants that underwent the same dehydration or desiccation treatment in the first and second experimental setups revealed that the increase in regeneration success after prolonged preculture on medium enriched with sucrose varied among genotypes (Table 2). The absolute value of this increase ranged as follows: (i) 6.2–27.4% for PVS A3-based dehydration; (ii) 14–30.6% for PVS3-based dehydration; (iii) 7.2–35.3% for 2 h desiccation; (iv) 23.5–36.1% for 3 h desiccation. The highest values of this increase were observed in genotypes that had previously shown a very low regeneration capacity or where it was not possible to regenerate shoots after cryo-preservation (‘Dragačevka’ and ‘Trnovača’). On the other hand, although this increase was evident, it was not significant for PVS A3-based dehydration in ‘Belošljiva’ and ‘Moravka’ or for PVS3-based dehydration in ‘Sitnica’.

Apart from the higher regrowth percentages, the regrowing shoots in this trial were more vigorous than the corresponding shoots regenerated in the first experimental setup using both the V cryo-plate method (Figure 3) and the D cryo-plate method (Figure 4).

### 2.2. Multiplication and Rooting Capacity of Shoot Regenerated from Cryo-Preserved Specimens

The effect of each step of both cryo-preservation protocols performed in the second experimental trial on the multiplication capacity of shoots regenerated from control and cryo-preserved specimens in the third subculture after regrowth is presented in Table 3. The steps included dissection, pre-growth, C4 loading treatment, dehydration (V cryo-plate), and desiccation (D cryo-plate) treatments, as well as freezing in LN.

Compared with dissection controls, the C4 loading treatment did not significantly affect any of the multiplication parameters of shoots in three genotypes (‘Moravka’, ‘Trnovača’, and ‘Crnošljiva’). However, in two genotypes (‘Belošljiva’ and ‘Cerovački Piskavac’), a significant increase was observed in the index of multiplication and length of axial shoots. Conversely, for C4 controls of three genotypes (‘Požegača’, ‘Dragačevka’, and ‘Sitnica’), the multiplication index was significantly lower than in corresponding dissection controls. Regarding the length of axial and lateral shoots, markedly lower values of both parameters were observed in ‘Dragačevka’, while plantlets of ‘Belošljiva’ and ‘Požegača’ (originated from C4 loading controls) had shorter lateral shoots compared with dissection controls.

Dehydration with VSs in the V cryo-plate protocol did not significantly affect the multiplication index of control shoots in the majority of genotypes, except in ‘Crnošljiva’ and ‘Cerovački Piskavac’, where a significant increase in this parameter was observed compared with dissection controls. Additionally, if varied, the length of axial and lateral shoots in dehydration controls was markedly higher than in plantlets regenerated from dissection controls. Conversely, a negative effect of dehydration was observed in ‘Požegača’, but only regarding the index of multiplication. On the other hand, a notable increase in multiplication index and/or shoot length was observed in shoots regenerated from cryo-preserved explants of all genotypes dehydrated with the PVS3 solution and in one genotype whose explants were dehydrated with the PVS A3 solution prior to exposure to LN (‘Belošljiva’). 

A striking contrast was observed when analyzing the results of shoots derived from control and cryo-preserved explants using the D cryo-plate method (Table 3) compared with the V cryo-plate method. Desiccation of explants for 2 or 3 h, followed by LN exposure, led to a notable reduction in the proliferation ability of regrowing plants in four genotypes out of the eight analyzed, as evident from the index of multiplication and/or length of shoots. Interestingly, only the control and cryo-preserved shoots of ‘Moravka’, as well as the cryo-preserved shoots of ‘Crnošljiva’, remained unaffected by desiccation. On the other hand, the cryo-preserved shoots of ‘Sitnica’ and ‘Cerovački Piskavac’ exhibited a significantly higher index of multiplication and length of axial shoots compared with corresponding dissection and desiccation controls.

Shoots regenerated from explants cryo-preserved using V cryo-plate and D cryo-plate methods displayed normal morphology and were well-developed (Figure 5). The shoots were a vibrant green and displayed vigorous growth with a healthy appearance. The shoots exhibited a well-developed structure with prominent apical meristems. Each shoot had multiple lateral buds, signifying active axillary branching and the potential for further shoot multiplication. The length of internodes was regular, contributing to a compact and robust growth habit.

Results on monitoring the rooting ability of shoots that underwent different treatments in the V cryo-plate and D cryo-plate protocols are presented in Table 4. Generally, pre-conditioning of explants on medium enriched with sucrose, as well as loading with C4 solution, did not affect the rooting rate of regrown shoots, except in ‘Dragačevka’ and ‘Crnošljiva’, where each treatment was followed by a gradual decrease in rooting efficiency. Regarding other parameters of the rooting ability of these shoots, if they were affected, the direction of that change depended on the monitored parameter and the genotype itself. 

In five genotypes, explants cryo-preserved using the V cryo-plate method and corresponding dehydration controls produced shoots that rooted just as efficiently as dissection controls. However, a significant decrease in the percentage of rooting was noticed in shoots originating from both control and cryo-preserved explants that were dehydrated with PVS A3 (in ‘Dragačevka’ and ‘Moravka’) or with PVS3 (in ‘Crnošljiva’).

Regarding the D cryo-plate method, desiccation had no impact on the rate at which shoots regenerated from cryo-preserved explants rooted, regardless of the treatment duration. The significant decrease in the percentage of rooting was only observed in shoots of ‘Požegača’, ‘Dragačevka’, and ‘Crnošljiva’, which were regenerated from desiccation controls. Other rooting parameters varied significantly in the majority of genotypes; the direction of variation depended on the specific genotype and the parameter being monitored (Table 4).

The roots of the in vitro plantlets regenerated from specimens cryo-preserved using both cryo-plate methods exhibited a healthy appearance with a bright white color. Notably, there was no presence of callus, indicating proper root differentiation. The root tips were well-defined and showed active growth. However, root hairs were not observed. The roots appeared to be free from contamination (Figure 6).

## 3. Discussion

Cryo-preservation allows long-term storage of plant material at ultra-low temperatures (LN), which suppresses all metabolic activities of cells [28] and ensures no genetic changes take place during storage [29]. Contemporary cryo-preservation techniques are based on vitrification; therefore, in most plant species, cell dehydration is required before freezing. This can be achieved by exposing samples to vitrification solutions and/or air dehydration at non-freezing temperatures. Consequently, the most critical step to achieving recovery growth in vitrification-based methods is the dehydration step, not the cooling step, as in cryo-preservation protocols based on freeze-induced dehydration [30,31].

The V cryo-plate method was developed with the aim of simplifying the cryo-preservation procedure by adhering explants to a reusable carrier with high thermal conductivity, which eliminates excessive manipulation and loss of plant material and enables ultra-high cooling and re-warming rates during the cryo-preservation process [23]. Furthermore, the timing of dehydration is more precisely controlled than in other vitrification techniques, therefore better survival and higher regrowth after cryopreservation are expected. Indeed, Vujović et al. [26] employed the V cryo-plate method for cryo-preservation of autochthonous genotype ‘Crvena Ranka’ and achieved 60% higher regrowth using PVS A3 and more than twice as high regeneration in explants dehydrated with PVS3 compared with the results of the droplet vitrification technique conducted under the same dehydration conditions. In this present study (first experimental setup), we also employed both 90% PVS2 (PVS A3) and PVS3 solutions, according to the recommendation that the type or modifications of PVSs should be chosen from cryo-protective solutions that give high regeneration in species of the same family [18]. Both VSs proved to be efficient for conservation of other *Prunus* genotypes, such as cherry plum and plum genotypes ‘Požegača’ and ‘Crvena Ranka’ [25,26]. In addition, a full-strength PVS2 solution, as widely used in most plant species, was utilized. According to Yamamoto et al. [32], almost all treatment conditions developed in any vitrification method might be applicable to the V cryo-plate method. With the exception of genotypes ‘Sitnica’ and ‘Požegača’, we did not conduct any previous cryo-preservation experiments for most of the tested genotypes. Therefore, durations of dehydration treatments were determined considering the results obtained in droplet vitrification experiments with ‘Sitnica’ [18] and cryo-plate experiments with ‘Požegača’ [25] and ‘Crvena Ranka’ [26]. As for PVS2-based dehydration under the described experimental conditions, the highest regrowth percentage (33.3%) was recorded for the longest treatment duration (40 min) in ‘Belošljiva’, followed by ‘Sitnica’, ‘Požegača’, ‘Moravka’, and ‘Cerovački Piskavac’, with regrowth ranging between 20% and 30%, which were poor results for successful conservation. Contrary to our results, cryo-protection in PVS2 solution enabled successful cryo-preservation of *Prunus* rootstocks using the slow cooling technique [33,34] and encapsulation dehydration [35], as well as *P. domestica* ‘Regina Claudia’ cryo-preserved by vitrification/one-step cooling and encapsulation–vitrification techniques [20]. Other than the toxicity of the vitrification solution used, there are a number of reasons for low regeneration of explants after cryo-preservation, such as insufficient osmotic adjustment of plant material, low penetration ability of cryo-protectants in plant tissues, excessive or insufficient dehydration, etc. [18]. Thus, it is necessary to check the regrowth of shoot tips following dehydration as such (dehydration controls), as the difference between the regeneration percentage of control and LN-treated specimens gives potential to improve regeneration by modification of PVS and/or treatment duration [36]. High regrowth of dehydration controls in ‘Belošljiva’, ‘Sitnica’, and ‘Moravka’ (between 90% and 100%) and the fact that regrowth of corresponding cryo-preserved explants increased with prolonged dehydration indicated that insufficient dehydration of explants could be the reason for low regeneration. Therefore, further extension of treatment could additionally increase regrowth in these genotypes. Similar results were obtained for PVS A3-based dehydration, with no significant differences compared with corresponding PVS2 treatments. The only exception was ‘Moravka’, where three-fold higher regrowth of cryo-preserved explants (62.5%) was obtained after 40 min dehydration with PVS A3. Regarding ‘Požegača’ and ‘Cerovački Piskavac’, a significant decrease in regeneration of control explants with prolonged dehydration with both PVS2 and PVS A3 suggested that optimal recovery after cryo-preservation could be achieved with an intermediate treatment duration (between 20 and 40 min). In fact, a 30 min treatment with PVS A3 solution at room temperature brought about regrowth of 44.6% in our previous research conducted with genotype ‘Požegača’ and higher than 55% with cherry plum [25]. Regarding ‘Crnošljiva’, a decrease in cryo-preservation success with prolonged dehydration without significant effect on regrowth of dehydration controls indicated that PVS treatment should be shorter than 20 min. Carrying out the PVS2 treatment at 0 °C could reduce the detrimental effects of the VS [37] and enable operational flexibility of the technique, as optimal growth was achieved over a much broader range of exposure durations [38]. On the other hand, genotypes ‘Dragačevka’ and ‘Trnovača’ displayed high sensitivity to both PVSs, with very low regrowth in dehydration controls and negligible regeneration after LN exposure. However, tolerance to PVS is acquired by optimizing not only the duration and temperature of exposure to PVS but also by preconditioning and loading treatments. Osmo-protection appears to be essential for inducing tolerance to dehydration; selecting the appropriate LS is a critical step when samples to be cryo-preserved are large and/or very sensitive to the chemical toxicity of the PVSs [39]. Although we used C4 LS, which was successfully applied in our previous experiments with *P. domestica* [26], further increases in sucrose concentration could significantly improve dehydration tolerance to PVSs in evaluated genotypes, as evidenced in Dalmatian chrysanthemum [23] and strawberry [32]. Unlike PVS2 solution and its variants, PVS3 can be used with plant materials that are very sensitive to biochemical toxicity and tolerant to osmotic stress [40]. In our experiments (first experimental setup), ‘Požegača’, ‘Dragačevka’, and ‘Trnovača’ displayed high sensitivity to the osmotic toxicity of PVS3 solution, with poor recovery of control explants and regrowth rates of cryo-preserved explants being mostly lower than 10%. Regarding ‘Moravka’, the results obtained (significantly higher regrowth with shorter treatment) suggest that sensitivity could be overcome by shortening the PVS3 exposure below 60 min. By contrast, genotypes ‘Belošljiva’, ‘Sitnica’, and ‘Crnošljiva’ were tolerant to this VS, as they showed high recovery of non-frozen controls. In addition, higher regrowth percentages in these genotypes were observed with a longer treatment duration, which indicated that optimal growth recovery might be achieved with additional PVS3 treatment extension. 

The D cryo-plate method that combined calcium–alginate encapsulation on a cryo-plate with dehydration [41] was developed with the aim of overcoming problems associated with sensitivity to PVSs and damage to plant material during chemical dehydration. This method proved to be successful in some representatives of the *Prunus* genus, such as cherry plum (77.5% regrowth after 3 h of desiccation) [25]. In the first experimental setup, regrowth higher than 40%, which is considered acceptable and reliable for gene banking purposes [42], was achieved in three (‘Belošljiva’, ‘Sitnica’, and ‘Požegača’) out of eight tested genotypes. The highest regrowth (40–65%) was noted in ‘Požegača’ and the results obtained were comparable to those previously reported in this genotype [25]. Regarding the other two genotypes, low recovery of control explants as well as a gradual decrease in regrowth of cryo-preserved ones with prolonged desiccation were similar to those obtained for ‘Crvena Ranka’ [26], which indicates that additional optimization of this step is required. Other genotypes displayed regrowth percentages of cryo-preserved specimens lower than 30% (three genotypes) or no regrowth at all (two genotypes), which was accomplished with low recovery of control explants as well. According to Niino et al. [41], if there was any survival after LN treatment, other than desiccation time, the concentration of sucrose in LS solution and duration of exposure to LS, as well as the concentration of sucrose in alginate gel, should be optimized. In addition, larger explants should be used to achieve more uniform dehydration [41].

Generally, the regrowth capacity of shoot tips cryo-preserved under the described experimental conditions in the first experimental setup depended on the cryo-preservation procedure used and exhibited a high level of genotype specificity. Genotypes that showed a high (100%) regrowth rate after loading treatment (both C4 and LS1) exhibited better regrowth in both dehydration/desiccation controls and corresponding cryo-preserved explants. Additionally, results obtained using the D cryo-plate method indicated that three genotypes with poor responses to LS1 solution displayed negligible or no regrowth after cryo-preservation. Therefore, selecting the appropriate LS is crucial for vitrification-based protocols, especially for plant species sensitive to the chemical toxicity of VSs [39], as well as physical dehydration. Also, pre-culture of shoot tips of plum accessions on agar-solidified medium with 0.3 M sucrose was insufficient conditioning for achieving high regrowth rates after cryo-preservation using both V cryo-plate and D cryo-plate methods. Pre-conditioning the donor plants through cold acclimation, exposure to high sugar levels, or anti-oxidants can enhance the physiological resilience of tissues against the stresses linked with cryo-preservation [43,44,45]. Cold hardening of donor plants has proven to be an essential step for successful cryo-preservation using a two-step freezing method in two inter-specific *Prunus* rootstocks, ‘Fereley-Jaspi’ and ‘Ferlenain-Plumina’ [33], and plum cultivars ‘Hamannova’, ‘Bílá Trnečka’, and ‘Chrudimer’ [46]. Although effective, the cold hardening treatment is time-consuming, typically taking more than 2–3 weeks of culture under low temperatures and reduced photo phase. It demands specific facilities, such as a cooled incubator, which could restrict the widespread application of cryo-preservation techniques. Moreover, the utilization of cold pre-treatment did not significantly improve the regrowth of cryo-preserved shoot tips in some fruit species, such as 12 *Vitis* species [47]. However, certain studies have indicated the potential of substituting the cold hardening of shoots of temperate fruit species, such as black currant [48] and sour cherry [49], with explant pre-culture on a medium containing a high concentration of soluble sugars. According to Bachiri et al. [50], similar to cold hardening, pre-culture leads to the accumulation of soluble sugars and proteins in cells, thereby enhancing their tolerance to cryo-preservation. Our results also confirmed that prolonged pre-culture of shoot tips on medium enriched with 0.5 M sucrose in total darkness at 4 °C (second experimental setup) can significantly increase cryo-preservation success in autochthonous plums using both V cryo-plate and D cryo-plate methods. In the V cryo-plate protocol, pre-culture of explants increases regrowth of dehydration controls in the majority of genotypes at 90% and above, which indicates that the applied treatment can successfully prevent explant injury caused by chemical dehydration with VSs. Consequently, post-cryopreservation regrowth percentages were also significantly improved and ranged between 45.7% and 68.4% in six genotypes. The results achieved in our study are comparable to those obtained with two *Prunus* rootstocks (69–74%) [33], using the two-step freezing method, and *P. avium* (50–80%) [51], using the one-step vitrification method, although they used cold-hardened shoots as the source of explants intended for cryo-preservation. Similar results were also obtained with two *P. cerasus* cultivars (41–63%) in experiments where the cold-hardening of mother plantlets was replaced with the pre-culture of shoot tips on medium enriched with glycerol and/or sucrose [49]. 

In the D cryo-plate protocol, the same pre-culture treatment together with a change in LS (C4 instead of LS1) also resulted in a significant improvement in the regrowth of desiccation controls across all genotypes. However, the regrowth rates achieved remained mostly below 90%, indicating that autochthonous plums are more sensitive to physical dehydration. Nevertheless, cryo-preserved explants of six genotypes showed high regrowth rates, ranging from 40.2% to 75%, which is also in agreement with results obtained in other *Prunus* species [33,49,51].

In two genotypes (‘Dragačevka’ and ‘Trnovača’) that showed very low regrowth using the V cryo-plate method or where regeneration of shoots was not possible after cryo-preservation using the D cryo-plate method (the first experimental setup), prolonged pre-culture of shoot tips in sucrose-enriched medium also significantly improved the success of cryo-preservation. However, the percentage of regrowth remained slightly below 40% or barely reached this value, which cannot be considered completely successful [52].

The ultimate goal of any cryo-preservation is not only the high regrowth success achieved with the optimized protocol, since the cryo-preservation method used may affect not only the cryo-preserved specimens but also the regenerated shoots [53]. For protocols that include an in vitro culture phase, it is critical to obtain true-to-type plantlets and actively growing cultures that are able to multiply and root. Continuous monitoring also provides valuable data on the efficiency of cryo-preservation protocols and allows the optimization of techniques to improve post-cryopreservation propagation rates. It helps researchers understand how various factors such as the choice of cryo-protectants, cooling rates, and storage duration affect the ability of regenerated shoots to multiply and root. 

In this present study, shoots of autochthonous plums cryo-preserved by the V cryo-plate method showed successful recovery (four genotypes) and even exceeded (four genotypes) the multiplication capacity of the dissection controls. This improvement was evident in the third subculture after regrowth, as indicated by the multiplication index and shoot length. Similar observations were reported for shoots of *P. cerasifera* and *P. domestica* L. regenerated from explants cryo-preserved by both the V cryo-plate and D cryo-plate methods compared with shoots recovered from untreated shoot tips [25]. On the other hand, monitoring of shoots recovered from explants cryo-preserved by the D cryo-plate in the currant study revealed that the ability to propagate was negatively affected in four genotypes, while two genotypes were unaffected and two performed better compared with the corresponding dissection controls. A significant decrease in the multiplication capacity of shoots regenerated from cryo-preserved explants using droplet vitrification was also reported in apple [54], while shoots of cherry plum were even unable to multiply over two successive subcultures after regrowth [55]. On the other hand, the multiplication step of olive cultures established from somatic embryos was not affected by cryo-preservation, although a significant interaction between genotype and cryo-preservation was found for shoot length during multiplication [56]. 

Assessing the rooting ability of the shoots regenerated from cryo-preserved specimens is crucial for successful plant recovery and acclimatization. Rooting ability reflects the plant’s capacity to establish a functional root system, enabling it to absorb water and nutrients from the soil. Without proper rooting, the regenerated plants may struggle to survive and grow after being reintroduced to their natural environment. Considering rooting rate as a parameter of rooting ability, shoots of all genotypes regenerated from explants cryo-preserved by the D cryo-plate method and of five genotypes cryo-preserved using the V cryo-plate method rooted with equal efficiency as dissection controls, while other rooting parameters varied significantly; the direction of variation depended on the genotype and the parameter being monitored. Bradaï and Sánchez-Romero [56] observed no significant differences in the parameters assessed in the rooting and acclimatization phases of olive plantlets regenerated from cryo-preserved embryogenic cultures. Nonetheless, slightly higher values of rooting percentage, number of roots per rooted shoot, and root length were achieved in shoots derived from cryo-preserved cultures of some lines compared with those obtained in control and unfrozen cultures. Therefore, this phenomenon, observed in plantlets recovered from cryo-preserved explants, is likely stress-induced and genotype-specific.

## 4. Materials and Methods

### 4.1. Plant Material

Eight autochthonous *P. domestica* genotypes ‘Belošljiva’, ‘Crnošljiva’, ‘Cerovački Piskavac’, ‘Dragačevka’, ‘Moravka’, ‘Požegača’, ‘Sitnica’, and ‘Trnovača’ were used for cryo-preservation experiments. Field-grown plants of these genotypes originating from heterogeneous local populations and various geographical locations in Serbia were used as sources of initial plant material for establishment of in vitro cultures. Virus-free mother plants were selected on the basis of 16 typical biological and production characteristics of the *P. domestica* autochthonous cultivars (true-to-type) described by Paunović [57]. Testing for the presence of seven viruses (plum pox virus, prune dwarf virus, *Prunus* necrotic ring spot virus, apple chlorotic leaf spot virus, apple mosaic virus, plum bark necrosis and stem pitting-associated virus, myrobalan latent ringspot virus) and ‘*Candidatus* Phytoplasma prunorum’ was performed by enzyme-linked immunosorbent assay and polymerase chain reaction. Aseptic cultures were established on Murashige and Skoog (MS) medium [58] containing 2 mg L^−1^ N6-benzyladenine (BA), 0.5 mg L^−1^ indole-3-butyric acid (IBA) and 0.1 mg L^−1^ gibberellic acid (GA_3_), 30 g L^−1^ sucrose, and 7 g L^−1^ agar (pH 5.7) according to the protocol previously described by Vujović et al. [25]. Following establishment of aseptic cultures and rosette initiation, genotypes were repeatedly subcultured on MS medium (pH 5.7) of constant plant growth regulator (PGR) composition: 1 mg L^−1^ BA, 0.1 mg L^−1^ 1-naphthaleneacetic acid (NAA) and 0.1 mg L^−1^ GA_3_. Subculturing was performed at 4 week intervals to obtain a sufficient number of mother stock axillary shoots for cryo-preservation experiments. Cultures were kept in a growth room at 23 ± 1 °C and a 16 h light/8 h dark photoperiod (light intensity, 41 μmol m^−2^ s^−1^). In order to simplify the procedure, MS medium of previously described PGR composition was used in all following steps of cryo-preservation protocols and for all tested genotypes.

### 4.2. Explant Dissection, Pregrowth, and Loading

Apical and axillary shoot tips (1.5 mm long) were dissected from the four-week-old in vitro shoots and pre-cultured for 1 day at 23 °C in the dark on solidified MS multiplication medium (pH 5.7) with 0.3 M sucrose (first experimental setup) or for 7 days on a medium containing 0.5 M sucrose in complete darkness at 4 °C (second experimental setup). After pre-culture, explants were carefully mounted on aluminum cryo-plates with 10 or 12 wells and embedded in alginate gel (2% (*w*/*v*) sodium alginate with 0.4 M sucrose in calcium-free MS medium polymerized for 20 min at room temperature using 0.1 M calcium chloride solution in MS medium with 0.4 M sucrose). The explants attached to cryo-plates were osmo-protected at 23 °C for 30 min in two different loading solutions comprising 2 M glycerol and 0.4 M sucrose (LS1 solution) [59] or 1.9 M glycerol and 0.5 M sucrose (C4 solution) [39] in liquid MS medium. The pH value was adjusted to 5.7 in both LSs. For each of the described steps, adequate non-frozen controls were included (dissection, pre-growth, and loading controls) as defined by Vujović et al. [26].

### 4.3. Cryopreservation Using Cryo-Plate Methods

#### 4.3.1. The First Experimental Setup

In the V cryo-plate protocol, following loading in C4 solution, explants (shortly pre-cultured at 23 °C on 0.3 M sucrose) were dehydrated using three different PVSs: the original PVS2 solution (13.7% sucrose, 30.0% glycerol, 15% ethylene glycol, and 15% dimethylsulfoxide) [60], the 90% PVS2 solution, PVS A3 solution (22.5% sucrose, 37.5% glycerol, 15% ethylene glycol, and 15% dimethylsulfoxide) [32], and the PVS3 solution (50% glycerol and 50% sucrose) [59]. The pH value was set to 5.7 for all PVSs. Dehydration was performed at 23 °C, while durations of treatments for each type of VS (20 and 40 min for PVS2 and PVS A3; 60 and 80 min for PVS3) were selected considering previously performed vitrification-based cryo-preservation experiments in different plum genotypes [22,25,26]. Re-warming of samples was performed at room temperature for 30 min in a 0.8 M sucrose solution in liquid MS medium with a pH of 5.7 [40]. Explants that were exposed to LS, dehydrated with VSs, and directly unloaded without immersion in LN were designated as dehydration controls (−LN).

In the D cryo-plate procedure, shoot tips were osmo-protected with LS1 solution and desiccated for 2, 2.5, or 3 h in closed 100 mL glass containers with over 40 g of silica gel at 23 °C in complete darkness. After dehydration, cryo-plates with adhering shoot tips were transferred to 2 mL uncapped cryo-tubes held on cryo-canes and directly plunged into LN, where they were kept for at least 1 h. Re-warming of samples was performed by rapid transfer of the aluminum cryo-plates into an unloading solution containing 0.8 M sucrose in liquid MS medium (pH 5.7) for 30 min. Desiccated but non-frozen shoot tips were used as desiccation controls (−LN).

#### 4.3.2. The Second Experimental Setup

In the second experiment, shoot tips that had been pre-cultured on a medium enriched with 0.5 M sucrose and loaded with C4 loading solution were used for both V cryo-plate and D cryo-plate cryo-preservation. Dehydration and desiccation treatments that generally yielded the best results using both methods in the first experimental setup were applied in a repeated trial. The experiment was designated as follows:Dehydration with PVS A3 for 40 min at 23 °C in ‘Belošljiva’, ‘Požegača’, ‘Moravka’, and ‘Dragačevka’;Dehydration with PVS3 for 80 min at 23 °C in ‘Sitnica’, ‘Crnošljiva’, ‘Cerovački Piskavac’, and ‘Trnovača’;Desiccation for 2 h over silica gel at 23 °C in ‘Belošljiva’, ‘Sitnica’, ‘Požegača’, ‘Dragačevka’, and ‘Trnovača’;Desiccation for 3 h over silica gel at 23 °C in ‘Moravka’, ‘Crnošljiva’, and ‘Cerovački Piskavac’.

All the following steps in cryo-preservation protocols were performed as previously described for the first experimental setup.

### 4.4. Growth Recovery and Statistical Analysis

Following unloading, both control and cryo-preserved explants were transferred to Petri dishes (5.5 cm diameter) on solidified regrowth medium (previously described medium for multiplication) and cultivated in the dark for 7 days at 23 ± 1 °C and then under standard conditions in the growth room (23 ± 1 °C; 16 h light/8 h dark photoperiod; 54 μmol m^−2^ s^−1^ light intensity). The effect of different treatments was determined by calculating the regrowth of explants. The development of shoot explants into viable shoots with developed leaves up to the sixth week was considered regrowth, while those that were green, swollen, or showed any sign of growth were considered to have survived but were not counted as regrowing (Figure 1a). Each experimental treatment was performed in three independent replicates, with 10–12 explants per replicate. For this purpose, 660–800 explants were isolated per genotype (22 treatments per genotype) in the first experimental setup; an additional 220 explants per genotype (6 treatments per genotype) were used for the second experimental setup. Statistical analysis was performed by one-way analysis of variance (ANOVA) and Duncan’s multiple range test (*p* < 0.05) for mean separation. Data presented as percentages were subjected to an arcsine transformation prior to analysis of variance.

### 4.5. Multiplication and Rooting Capacity of Shoots Regenerated from Cryo-Preserved Specimens

Following regrowth in the second experimental setup, shoots of each genotype originating from dissection, pre-growth, and loading controls, as well as those originating from dehydration and desiccation treatments (both control and cryo-preserved), were separately transferred onto the MS multiplication medium. The multiplication index and length of axial and lateral shoots were monitored in the third subculture after regrowth (four-week culture interval). In the next (fourth) subculture, the shoots were rooted in vitro on the MS medium with mineral salts reduced to ½-strength, the organic complex unchanged, and the following hormonal composition previously found to be the most suitable for each genotype:1 mg L^−1^ IBA and 0.1 mg L^−1^ GA_3_ in ‘Belošljiva’, ‘Trnovača’, ‘Dragačevka’, and ‘Cerovački Piskavac’;1 mg L^−1^ NAA and 0.1 mg L^−1^ GA_3_ in ‘Sitnica’, ‘Požegača’, ‘Crnošljiva’, and ‘Moravka’.

The monitoring of rooting capacity was performed after four weeks and included the following parameters: percentage of rooted plantlets, number and length of roots, and length of rooted shoots. 

All parameters were measured in at least 45 randomly selected plantlets of different origin in each genotype (3 replicates with at least 15 plantlets per replicate). All data were analyzed by ANOVA, followed by the Duncan’s multiple range test, at *p* ≤ 0.05.

## 5. Conclusions

The results obtained in this study clearly indicate that autochthonous plum genotypes/cultivars react differently not only to the cryo-preservation method used (V or D cryo-plate) but also to the dehydration and desiccation treatments applied. Indeed, despite the intention to develop a protocol that would be widely applicable to different genotypes within the same species, some studies have shown significant variation regarding cryo-preservation ability between different cultivars belonging to the same species [42,61], which is in accordance with observations in our research.

Regarding all analyzed genotypes, it is clear that pre-conditioning plant material intended for cryo-preservation is necessary for the successful cryo-preservation of autochthonous plums using vitrification procedures. Notably, a significant improvement in regrowth success for all genotypes was achieved by pre-culturing shoot tips for 7 days on a medium containing 0.5 M sucrose in complete darkness at 4 °C. In six out of the eight analyzed genotypes, post-cryopreservation regrowth percentages ranged between 45.7% and 68.4% using the V cryo-plate method and between 40.2% and 75% using the D cryo-plate method. In two genotypes, regrowth percentages remained slightly below 40% or barely reached that level, which might not be considered entirely successful.

This research also highlights the significance of monitoring the multiplication and rooting capabilities of shoots regenerated from cryo-preserved explants as a fundamental aspect of cryo-preservation research and plant conservation efforts. Our study demonstrates that, across all genotypes, shoots regenerated from explants cryo-preserved through refined protocols for both the V cryo-plate and D cryo-plate methods have successfully retained and even exceeded the capacity for multiplication and rooting compared with control (non-frozen) shoots across successive subcultures following post-cryopreservation regrowth.

Although no visible morphological variations were found in shoots regenerated from cryo-preserved explants with respect to control shoots, these plants will be further monitored in the open field for any changes in important agronomic traits.

## Figures and Tables

**Figure 1 plants-12-03108-f001:**
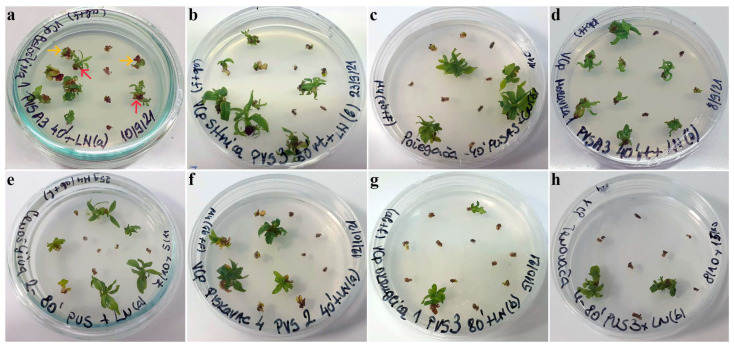
Regrowth of shoot tips of autochthonous plum genotypes cryo-preserved using V cryo-plate method (1st experimental setup). Explants were precultured for 1 day on 0.3 M sucrose at 23 °C and dehydrated with: (**a**) PVS A3 for 40 min, ‘Belošljiva’ (the red arrows indicate regrowth, while the yellow arrows show explants that are considered as having survived but not regrown); (**b**) PVS3 for 80 min, ‘Sitnica’; (**c**) PVS A3 for 40 min, ‘Požegača’; (**d**) PVS A3 for 40 min, ‘Moravka’; (**e**) PVS3 for 80 min, ‘Crnošljiva’; (**f**) PVS2 for 40 min, ‘Cerovački Piskavac’; (**g**) PVS3 for 80 min, ‘Dragačevka’; (**h**) PVS3 for 80 min, ‘Trnovača’.

**Figure 2 plants-12-03108-f002:**
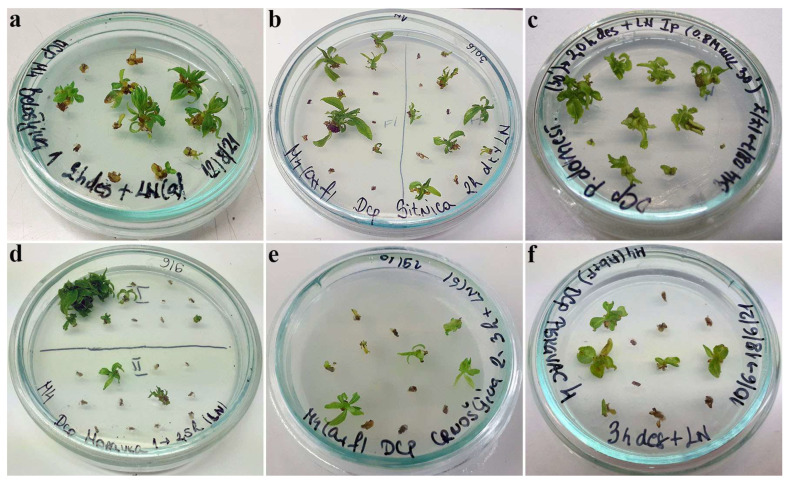
Regrowth of shoot tips of autochthonous plum genotypes cryo-preserved using D cryo-plate method (1st experimental setup). Explants were pre-cultured for 1 day on 0.3 M sucrose at 23 °C and desiccated for 2 h: (**a**) ‘Belošljiva’, (**b**) ‘Sitnica’, (**c**) ‘Požegača’; for 2.5 h: (**d**) ‘Moravka’; for 3 h: (**e**) ‘Crnošljiva’ and (**f**) ‘Cerovački Piskavac’.

**Figure 3 plants-12-03108-f003:**
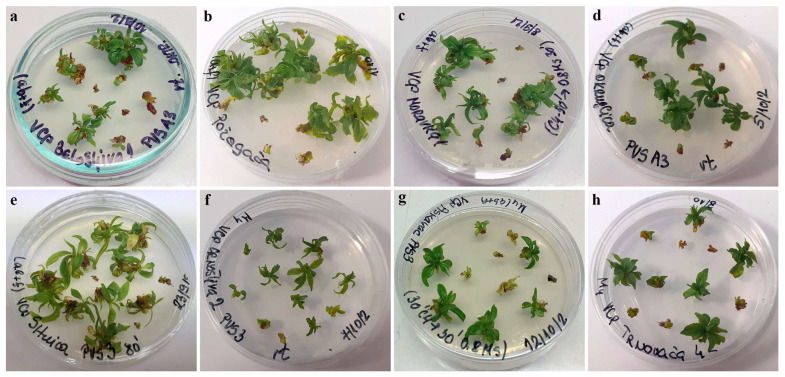
Regrowth of shoot tips of autochthonous plum genotypes cryo-preserved using V cryo-plate method (2nd experimental setup). Explants were pre-cultured for 7 days on 0.5 M sucrose at 4 °C and dehydrated with: PVS A3 for 40 min, (**a**) ‘Belošljiva’, (**b**) ‘Požegača’, (**c**) ‘Moravka’, and (**d**) ‘Dragačevka’; PVS3 for 80 min, (**e**) ‘Sitnica’, (**f**) ‘Crnošljiva’, (**g**) ‘Cerovački Piskavac’, and (**h**) ‘Trnovača’.

**Figure 4 plants-12-03108-f004:**
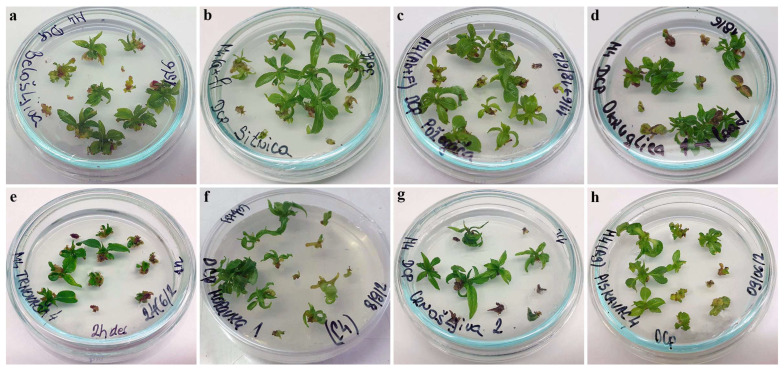
Regrowth of shoot tips of autochthonous plum genotypes cryo-preserved using D cryo-plate method (2nd experimental setup). Explants were precultured for 7 days on 0.5 M sucrose at 4 °C and desiccated for: 2 h, (**a**) ‘Belošljiva’, (**b**) ‘Sitnica’, (**c**) ‘Požegača’, (**d**) ‘Dragačevka’, and (**e**) ‘Trnovača’; 3 h, (**f**) ‘Moravka’, (**g**) ‘Crnošljiva’, and (**h**) ‘Cerovački Piskavac’.

**Figure 5 plants-12-03108-f005:**
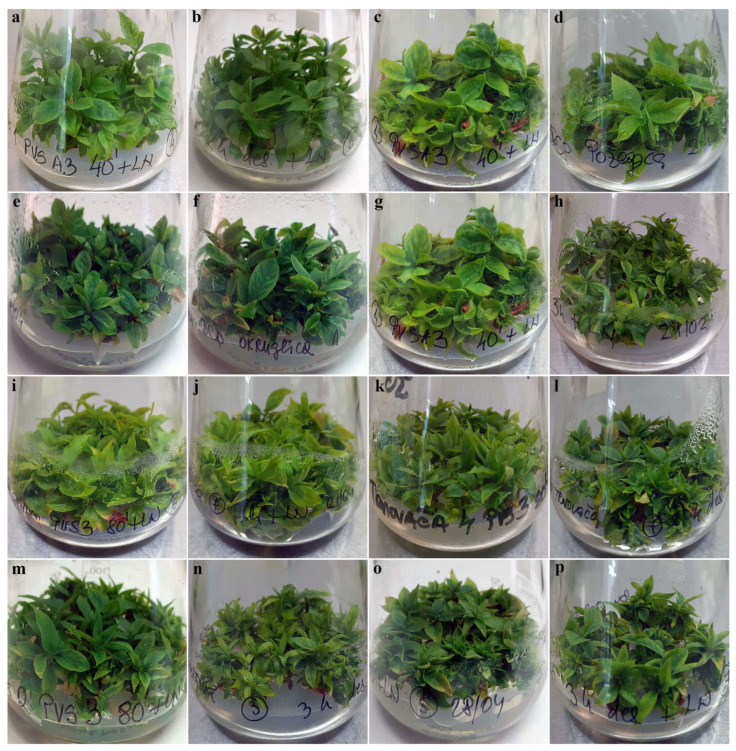
Multiplication in vitro of autochthonous plums regenerated from explants cryo-preserved using V cryo-plate and D cryo-plate methods (2nd experimental setup). Shoots originated from explants dehydrated with PVS A3 for 40 min or desiccated for 2 h, accordingly: (**a**,**b**) ‘Belošljiva’, (**c**,**d**) ‘Požegača’, and (**e**,**f**) ‘Dragačevka’. Shoots originated from explants dehydrated with PVS A3 for 40 min or desiccated for 3 h, accordingly: (**g**,**h**) ‘Moravka’. Shoots originated from explants dehydrated with PVS3 for 80 min or desiccated for 2 h, accordingly: (**i**,**j**) ‘Sitnica’ and (**k**,**l**) ‘Trnovača’. Shoots originated from explants dehydrated with PVS3 for 80 min or desiccated for 3 h, accordingly: (**m**,**n**) ‘Crnošljiva’ and (**o**,**p**) ‘Cerovački Piskavac‘.

**Figure 6 plants-12-03108-f006:**
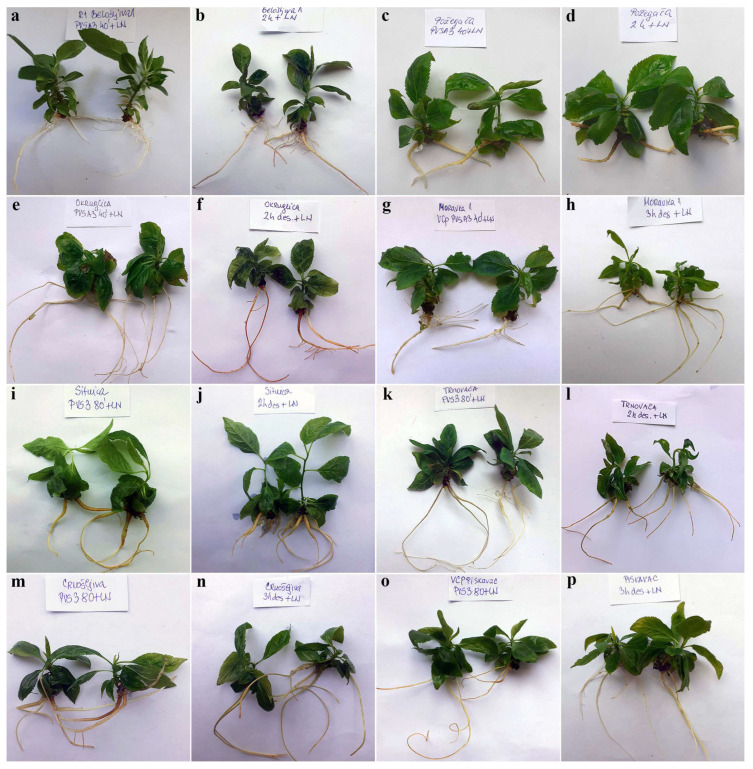
In vitro rooting of autochthonous plums regenerated from explants cryo-preserved using V cryo-plate and D cryo-plate methods (2nd experimental setup). Shoots originated from explants dehydrated with PVS A3 for 40 min or desiccated for 2 h, accordingly: (**a**,**b**) ‘Belošljiva’, (**c**,**d**) ‘Požegača’, and (**e**,**f**) ‘Dragačevka’. Shoots originated from explants dehydrated with PVS A3 for 40 min or desiccated for 3 h, accordingly: (**g**,**h**) ‘Moravka’. Shoots originated from explants dehydrated with PVS3 for 80 min or desiccated for 2 h, accordingly: (**i**,**j**) ‘Sitnica’ and (**k**,**l**) ‘Trnovača’. Shoots originated from explants dehydrated with PVS3 for 80 min or desiccated for 3 h, accordingly: (**m**,**n**) ‘Crnošljiva’ and (**o**,**p**) ‘Cerovački Piskavac’.

**Table 1 plants-12-03108-t001:** Regrowth (%) of control (−LN) and cryo-preserved (+LN) shoot tips of autochthonous plum genotypes using the V cryo-plate and D cryo-plate methods.

Treatment	Regrowth (%)															
	‘Belošljiva’		‘Sitnica’		‘Požegača’		‘Moravka’		‘Crnošljiva’		‘Cerovački Piskavac’		‘Dragačevka’		‘Trnovača’	
	−LN	+LN	−LN	+LN	−LN	+LN	−LN	+LN	−LN	+LN	−LN	+LN	−LN	+LN	−LN	+LN
Dissection control	100.0 ^a^	-	100.0 ^a^	-	100.0 ^a^	-	100.0 ^a^	-	100.0 ^a^	-	100.0 ^a^	-	100.0 ^a^	-	90.0 ^b^	-
Pre-growth control	100.0 ^a^	-	100.0 ^a^	-	100.0 ^a^	-	100.0 ^a^	-	100.0 ^a^	-	100.0 ^a^	-	100.0 ^a^	-	90.0 ^b^	-
LS1 control	100.0 ^a^	-	100.0 ^a^	-	100.0 ^a^	-	100.0 ^a^	-	50.0 ^c^	-	100.0 ^a^	-	60.0 ^de^	-	50.0 ^c^	-
C4 control	100.0 ^a^	-	100.0 ^a^	-	100.0 ^a^	-	100.0 ^a^	-	100.0 ^a^	-	90.0 ^b^	-	80.0 ^bc^	-	100.0 ^a^	-
C4–PVS2 20 min	90.0 ^b^	4.2 ^j^	100.0 ^a^	4.2 ^e^	60.0 ^def^	12.9 ^kl^	100.0 ^a^	0.0 ^i^	100.0 ^a^	17.2 ^ef^	60.0 ^cd^	0.0 ^h^	40.0 ^fg^	4.2 ^i^	20.0 ^de^	0.0 ^g^
C4–PVS2 40 min	100.0 ^a^	33.3 ^gh^	100.0 ^a^	25.0 ^cd^	20.0 ^jk^	25.0 ^ij^	90.0 ^b^	20.8 ^fg^	100.0 ^a^	0.0 ^g^	30.0 ^ef^	30.0 ^ef^	30.0 ^g^	0.0 ^i^	20.0 ^de^	8.3 ^ef^
C4–PVS A3 20 min	90.0 ^b^	17.4 ^i^	100.0 ^a^	12.5 ^d^	90.0 ^b^	9.7 ^l^	100.0 ^a^	0.0 ^i^	100.0 ^a^	30.0 ^de^	80.0 ^bc^	8.3 ^gh^	90.0 ^b^	0.0 ^i^	30.0 ^cd^	8.3 ^ef^
C4–PVS A3 40 min	80.0 ^bc^	37.5 ^fgh^	50.0 ^b^	16.7 ^d^	60.0 ^def^	33.3 ^hi^	80.0 ^c^	62.5 ^de^	100.0 ^a^	0.0 ^g^	30.0 ^ef^	16.7 ^fg^	80.0 ^bc^	12.5 ^h^	10.0 ^def^	4.2 ^fg^
C4–PVS3 60 min	90.0 ^b^	25.0 ^hi^	100.0 ^a^	37.5 ^bc^	50.0 ^fg^	4.2 ^m^	50.0 ^e^	48.1 ^e^	100.0 ^a^	25.0 ^def^	60.0 ^cd^	20.8 ^ef^	70.0 ^cd^	0.0 ^i^	10.0 ^def^	4.2 ^fg^
C4–PVS3 80 min	70.0 ^cd^	29.2 ^ghi^	100.0 ^a^	45.8 ^b^	20.0 ^jk^	8.3 ^l^	30.0 ^f^	13.3 ^gh^	88.9 ^b^	42.9 ^cd^	30.0 ^ef^	25.0 ^ef^	50.0 ^ef^	16.7 ^h^	11.1 ^def^	11.1 ^def^
LS1–2 h desiccation	60.0 ^de^	41.7 ^fg^	50.0 ^b^	41.6 ^b^	80.0 ^c^	65.0 ^de^	70.0 ^cd^	10.0 ^h^	30.0 ^de^	12.5 ^f^	40.0 ^de^	0.0 ^h^	10.0 ^h^	0.0 ^i^	0.0 ^g^	0.0 ^g^
LS1–2.5 h desiccation	50.0 ^ef^	29.2 ^ghi^	50.0 ^b^	39.4 ^b^	70.0 ^cd^	55.0 ^ef^	60.0 ^de^	29.2 ^f^	20.0 ^ef^	12.5 ^f^	30.0 ^ef^	8.3 ^gh^	10.0 ^h^	0.0 ^i^	8.3 ^ef^	4.2 ^fg^
LS1–3 h desiccation	30.0 ^ghi^	33.3 ^gh^	20.0 ^d^	25.0 ^cd^	63.3 ^def^	40.0 ^gh^	50.0 ^e^	25.0 ^f^	20.0 ^ef^	16.7 ^ef^	30.0 ^ef^	25.0 ^ef^	0.0 ^i^	0.0 ^i^	0.0 ^g^	0.0 ^g^
Significance	*p* ≤ 0.05		*p* ≤ 0.05		*p* ≤ 0.05		*p* ≤ 0.05		*p* ≤ 0.05		*p* ≤ 0.05		*p* ≤ 0.05		*p* ≤ 0.05	

Mean values of regrowth in each genotype (arranged in column pairs and labelled with the same small letter in superscript) were not significantly different according to Duncan’s multiple range test. Regrowth was determined six weeks after transfer of explants on medium for regrowth. LS1—loading solution comprising 2 M glycerol and 0.4 M sucrose; C4—loading solution comprising 1.9 M glycerol and 0.5 M sucrose. PVS2—plant vitrification solution comprising 13.7% sucrose, 30.0% glycerol, 15% ethylene glycol, and 15% dimethylsulfoxide; PVS A3—90% PVS2 solution (22.5% sucrose, 37.5% glycerol, 15% ethylene glycol, and 15% dimethylsulfoxide); PVS3—plant vitrification solution comprising 50% glycerol and 50% sucrose.

**Table 2 plants-12-03108-t002:** Effect of preculture of shoot tips and type of loading solution on regrowth (%) of control (−LN) and cryo-preserved (+LN) shoot tips of autochthonous plum genotypes using the V cryo-plate and D cryo-plate methods.

Treatment	Regrowth (%)															
	‘Belošljiva’		‘Sitnica’		‘Požegača’		‘Moravka’		‘Crnošljiva’		‘Cerovački Piskavac’		‘Dragačevka’		‘Trnovača’	
	−LN	+LN	−LN	+LN	−LN	+LN	−LN	+LN	−LN	+LN	−LN	+LN	−LN	+LN	−LN	+LN
Dissection control	100.0 ^a^	-	100.0 ^a^	-	100.0 ^a^	-	100.0 ^a^	-	100.0 ^a^	-	100.0 ^a^	-	100.0 ^a^	-	90.0 ^b^	-
Pre-growth control *	100.0 ^a^	-	100.0 ^a^	-	100.0 ^a^	-	100.0 ^a^	-	100.0 ^a^	-	100.0 ^a^	-	100.0 ^a^	-	100.0 ^a^	-
LS1 control	100.0 ^a^	-	100.0 ^a^	-	100.0 ^a^	-	100.0 ^a^	-	50.0 ^bcd^	-	100.0 ^a^	-	60.0 ^c^	-	50.0 ^d^	-
C4 control *	100.0 ^a^	-	100.0 ^a^	-	100.0 ^a^	-	100.0 ^a^	-	100.0 ^a^	-	100.0 ^a^	-	100.0 ^a^	-	100.0 ^a^	-
C4–PVS A3 40 min	80.0 ^b^	37.5 ^d^	NC.		60.0 ^ef^	33.3 ^g^	80.0 ^b^	62.5 ^c^	NC.		NC.		80.0 ^b^	12.5 ^e^	NC.	
C4–PVS A3 40 min *	100.0 ^a^	54.2 ^cd^			90.0 ^b^	58.3 ^f^	100.0 ^a^	68.7 ^c^					100.0 ^a^	39.9 ^d^		
C4–PVS3 80 min	NC.		100.0 ^a^	45.8 ^cd^	NC.		NC.		88.9 ^a^	42.9 ^cd^	30.0 ^d^	25.0 ^d^	NC.		11.1 ^g^	11.1 ^g^
C4–PVS3 80 min *			100.0 ^a^	59.8 ^c^					100.0 ^a^	68.4 ^b^	80.0 ^b^	45.7 ^c^			70.0 ^c^	41.7 ^e^
LS1–2 h desiccation	60.0 ^c^	41.7 ^d^	50.0 ^cd^	41.6 ^d^	80.0 ^c^	65.0 ^e^	NC.		NC.		NC.		10.0 ^e^	0.0 ^f^	0.0 ^h^	0.0 ^h^
C4–2 h desiccation *	96.7 ^a^	66.7 ^bc^	83.3 ^b^	75.0 ^b^	100.0 ^a^	72.2 ^d^							76.7 ^b^	35.3 ^d^	63.3 ^c^	30.6 ^f^
LS1–3 h desiccation	NC.		NC.		NC.		50.0 ^d^	25.0 ^e^	20.0 ^ef^	16.7 ^f^	30.0 ^d^	25.0 ^d^	NC.		NC.	
C4–3 h desiccation *							90.0 ^a^	61.1 ^c^	63.3 ^bc^	40.2 ^de^	86.7 ^b^	58.3 ^c^				
	*p* ≤ 0.05		*p* ≤ 0.05		*p* ≤ 0.05		*p* ≤ 0.05		*p* ≤ 0.05		*p* ≤ 0.05		*p* ≤ 0.05		*p* ≤ 0.05	

Mean values of regrowth in each genotype (arranged in column pairs and labelled with the same small letter in superscript) were not significantly different according to Duncan’s multiple range test. Regrowth was determined six weeks after transfer of explants on medium for regrowth. LS1—loading solution comprising 2 M glycerol and 0.4 M sucrose; C4—loading solution comprising 1.9 M glycerol and 0.5 M sucrose. PVS2—plant vitrification solution comprising 13.7% sucrose, 30.0% glycerol, 15% ethylene glycol, and 15% dimethylsulfoxide; PVS A3—90% PVS2 solution (22.5% sucrose, 37.5% glycerol, 15% ethylene glycol, and 15% dimethylsulfoxide); PVS3—plant vitrification solution comprising 50% glycerol and 50% sucrose. * Refers to shoot tips precultured for 7 days on a medium containing 0.5 M sucrose in complete darkness at 4 °C. In treatments that are not followed by an asterisk, shoot tips were pre-cultured for 1 day on a medium containing 0.3 M sucrose in complete darkness at +23 °C. NC.—not compared (as the second experiment did not perform).

**Table 3 plants-12-03108-t003:** Parameters of multiplication of shoots of autochthonous plums regenerated from cryo-preserved explants in comparison with those regenerated from control explants (3rd subculture).

Genotype/ Parameter	Treatment						
	Dissection Control	Pre-growth Control	C4 Control	C4—PVSA3 40 min −LN	C4—PVSA3 40 min +LN	C4—2 h Desic. −LN	C4—2 h Desic. +LN
**‘Belošljiva’**							
Index of multiplication	4.6 ^bc^	4.2 ^c^	5.3 ^a^	5.0 ^ab^	5.3 ^a^	4.5 ^bc^	4.2 ^c^
Length of axial shoot (mm)	13.2 ^b^	12.3 ^c^	14.4 ^a^	14.3 ^a^	12.4 ^bc^	11.0 ^d^	11.8 ^cd^
Length of lateral shoot (mm)	7.5 ^b^	7.3 ^bc^	6.9 ^c^	7.2 ^bc^	8.2 ^a^	7.5 ^b^	7.3 ^bc^
**‘Požegača’**							
Index of multiplication	4.4 ^a^	3.6 ^bc^	3.9 ^b^	3.6 ^b^	4.3 ^a^	3.0 ^d^	3.6 ^bc^
Length of axial shoot (mm)	11.0 ^a^	11.2 ^a^	11.6 ^a^	11.3 ^a^	11.5 ^a^	10.3 ^b^	11.6 ^a^
Length of lateral shoot (mm)	7.6 ^a^	7.6 ^a^	6.6 ^b^	6.3 ^b^	6.5 ^b^	6.4 ^b^	6.4 ^b^
**‘Dragačevka’**							
Index of multiplication	4.6 ^a^	4.8 ^a^	3.9 ^b^	4.9 ^a^	5.1 ^a^	3.2 ^c^	3.4 ^bc^
Length of axial shoot (mm)	12.1 ^a^	1.14 ^b^	1.11 ^b^	1.22 ^a^	1.24 ^a^	1.03 ^c^	1.01 ^c^
Length of lateral shoot (mm)	9.1 ^b^	9.9 ^ab^	7.5 ^c^	10.2 ^a^	10.8 ^a^	6.4 ^d^	6.4 ^d^
**Genotype/** **Parameter**	**Dissection control**	**Pre-growth control**	**C4** **control**	**C4—PVSA3 40 min − LN**	**C4—PVSA3 40 min + LN**	**C4—3 h desic. − LN**	**C4—3 h desic. + LN**
**‘Moravka’**							
Index of multiplication	3.8 ^abc^	4.2 ^a^	4.1 ^ab^	3.8 ^abc^	3.6 ^bc^	3.4 ^c^	4.1 ^ab^
Length of axial shoot (mm)	10.8	10.2	11.4	11.2	10.7	11.0	10.6
Length of lateral shoot (mm)	6.5 ^bc^	7.1 ^ab^	7.1 ^ab^	7.2 ^a^	6.8 ^abc^	6.9 ^abc^	6.3 ^c^
**Genotype/** **Parameter**	**Dissection control**	**Pre-growth control**	**C4** **control**	**C4—PVS3 80 min − LN**	**C4—PVS3 80 min + LN**	**C4—2 h desic. − LN**	**C4—2 h desic. + LN**
**‘Sitnica’**							
Index of multiplication	4.1 ^c^	4.3 ^c^	3.5 ^d^	4.4 ^c^	4.8 ^b^	3.8 ^d^	5.1 ^a^
Length of axial shoot (mm)	12.1 ^cd^	14.7 ^a^	12.7 ^bcd^	13.0 ^bc^	13.2 ^b^	12.0 ^d^	14.2 ^a^
Length of lateral shoot (mm)	6.5 ^c^	7.4 ^b^	6.8 ^c^	7.4 ^b^	7.2 ^b^	6.7 ^c^	8.0 ^a^
**‘Trnovača’**							
Index of multiplication	3.9 ^b^	3.9 ^b^	4.2 ^ab^	4.1 ^ab^	4.3 ^a^	3.2 ^c^	3.2 ^c^
Length of axial shoot (mm)	12.0 ^b^	13.2 ^a^	11.8 ^bc^	11.6 ^bcd^	10.8 ^d^	11.9 ^b^	10.9 ^cd^
Length of lateral shoot (mm)	6.8 ^abc^	6.6 ^cd^	7.2 ^a^	7.1 ^ab^	6.3 ^d^	6.6 ^cd^	6.7 ^bc^
**Genotype/** **Parameter**	**Dissection control**	**Pre-growth control**	**C4** **control**	**C4—PVS3 80 min − LN**	**C4—PVS3 80 min + LN**	**C4—3 h desic. − LN**	**C4—3 h desic. + LN**
**‘Crnošljiva’**							
Index of multiplication	4.2 ^c^	4.1 ^c^	3.9 ^cd^	5.1 ^b^	6.5 ^a^	3.5 ^d^	4.3 ^c^
Length of axial shoot (mm)	11.8 ^ab^	11.3 ^bc^	12.2 ^a^	12.6 ^a^	11.9 ^ab^	10.8 ^c^	11.2 ^bc^
Length of lateral shoot (mm)	7.3 ^b^	6.9 ^b^	7.1 ^b^	9.5 ^a^	9.2 ^a^	6.0 ^c^	7.3 ^b^
**‘Cerovački Piskavac’**							
Index of multiplication	3.2 ^de^	3.4 ^cd^	3.8 ^b^	4.7 ^a^	4.8 ^a^	3.1 ^e^	3.7 ^bc^
Length of axial shoot (mm)	9.6 ^d^	10.0 ^cd^	11.7 ^a^	10.8 ^bc^	11.0 ^ab^	10.4 ^bcd^	10.6 ^bc^
Length of lateral shoot (mm)	6.6 ^bcd^	6.4 ^cd^	6.9 ^b^	6.7 ^bc^	7.6 ^a^	6.2 ^d^	6.6 ^bcd^

For each genotype, mean values of each parameter (multiplication index, length of axial shoot, and length of lateral shoot) within a single row (labelled with the same small letter in the superscript) were not significantly different according to Duncan’s multiple range test (*p* ≤ 0.05). If the values are not followed with small letters at all, they were not significantly different. +LN—cryo-preserved explants; −LN—non-cryo-preserved explants. All shoots, except those regenerated from dissection controls, were originated from specimens that were pre-cultured for 7 days on a medium containing 0.5 M sucrose in complete darkness at 4 °C.

**Table 4 plants-12-03108-t004:** Parameters of rooting of shoots of autochthonous plums regenerated from cryo-preserved explants in comparison with those regenerated from control explants (4th subculture).

Genotype/ Parameter	Treatment						
	Dissection Control	Pre-growth Control	C4 Control	C4—PVSA3 40 min −LN	C4—PVSA3 40 min +LN	C4—2 h Desic. −LN	C4—2 h Desic. +LN
**‘Belošljiva’**							
Rooting rate (%)	93.3	93.3	82.2	73.3	80.0	86.7	90.0
No. of roots	3.3 ^a^	2.8 ^a^	2.1 ^b^	1.4 ^c^	1.9 ^bc^	1.6 ^bc^	2.8 ^a^
Root length (mm)	45.3 ^b^	44.9 ^b^	37.9 ^c^	48.1 ^ab^	44.3 ^b^	52.8 ^a^	45.6 ^b^
Shoot height (mm)	15.5 ^b^	16.1 ^b^	20.5 ^a^	19.8 ^a^	21.2 ^a^	19.8 ^a^	19.7 ^a^
**‘Požegača’**							
Rooting rate (%)	77.8 ^ab^	82.2 ^a^	71.1 ^ab^	43.3 ^ab^	71.1 ^ab^	53.3 ^c^	62.2 ^bc^
No. of roots	2.2 ^a^	1.9 ^ab^	2.3 ^a^	1.4 ^bc^	2.0 ^a^	1.4 ^c^	1.2 ^c^
Root length (mm)	19.3 ^bc^	15.6 ^c^	31.9 ^a^	34.8 ^a^	22.3 ^b^	34.8 ^a^	23.7 ^b^
Shoot height (mm)	13.7 ^abc^	15.3 ^ab^	13.4 ^abc^	15.8 ^a^	16.0 ^a^	12.9 ^bc^	12.2 ^c^
**‘Dragačevka’**							
Rooting rate (%)	95.6 ^a^	75.6 ^b^	66.7 ^b^	66.7 ^b^	75.6 ^b^	77.8 ^b^	95.6 ^a^
No. of roots	2.6 ^a^	2.7 ^a^	2.1 ^b^	2.6 ^a^	2.8 ^a^	1.6 ^c^	2.0 ^bc^
Root length (mm)	45.9 ^d^	66.3 ^ab^	35.0 ^e^	72.5 ^a^	54.9 ^c^	61.9 ^b^	61. ^bc^
Shoot height (mm)	14.9	13.6	13.7	14.8	15.0	14.8	15.4
**Genotype/** **Parameter**	**Dissection control**	**Pre-growth control**	**C4** **control**	**C4—PVSA3 40 min − LN**	**C4—PVSA3 40 min + LN**	**C4—3 h desic. − LN**	**C4—3 h desic. + LN**
**‘Moravka’**							
Rooting rate (%)	86.7 ^ab^	95.6 ^a^	76.7 ^bc^	73.3 ^c^	68.9 ^c^	84.5 ^ab^	97.7 ^a^
No. of roots	2.2 ^bc^	2.0 ^c^	1.6 ^d^	1.6 ^d^	1.3 ^d^	2.8 ^a^	2.5 ^ab^
Root length (mm)	43.0 ^a^	36.5 ^ab^	27.3 ^c^	26.0 ^c^	26.2 ^c^	36.9 ^ab^	35.2 ^b^
Shoot height (mm)	21.8 ^a^	18.4 ^b^	22.1 ^a^	23.1 ^a^	21.8 ^a^	22.3 ^a^	19.5 ^ab^
**Genotype/** **Parameter**	**Dissection control**	**Pre-growth control**	**C4** **control**	**C4—PVS3 80 min − LN**	**C4—PVS3 80 min + LN**	**C4—2 h desic. − LN**	**C4—2 h desic. + LN**
**‘Sitnica’**							
Rooting rate (%)	71.1	84.3	82.2	86.7	50	73.33	64.45
No. of roots	1.4 ^c^	2.0 ^b^	2.1 ^b^	2.7 ^a^	2.6 ^a^	1.4 ^c^	1.7 ^bc^
Root length (mm)	64.8 ^a^	51.0 ^b^	47.4 ^bc^	10.4 ^c^	45.0 ^bc^	70.2 ^a^	45.2 ^bc^
Shoot height (mm)	15.0 ^ab^	14.2 ^bc^	14.1 ^bc^	16.2 ^a^	16.0 ^a^	13.1 ^cd^	12.3 ^d^
**‘Trnovača’**							
Rooting rate (%)	84.5 ^ab^	93.3 ^a^	82.2 ^ab^	80.0 ^ab^	84.5 ^ab^	77.8 ^b^	82.2 ^ab^
No. of roots	2.9 ^b^	4.3 ^a^	2.7 ^bc^	2.3 ^bc^	2.8^bc^	2.0 ^c^	2.2 ^bc^
Root length (mm)	40.8 ^bc^	36.7 ^c^	44.9 ^ab^	40.7 ^bc^	45.6 ^ab^	49.3 ^a^	46.9 ^a^
Shoot height (mm)	14.1	15.0	14.8	13.8	14.4	14.5	14.1
**Genotype/** **Parameter**	**Dissection control**	**Pre-growth control**	**C4** **control**	**C4—PVS3 80 min − LN**	**C4—PVS3 80 min + LN**	**C4—3 h desic. −LN**	**C4—3 h desic. +LN**
**‘Crnošljiva’**							
Rooting rate (%)	91.1 ^a^	73.3 ^bc^	66.7 ^c^	71.1 ^bc^	75.6 ^bc^	71.1^bc^	84.5 ^ab^
No. of roots	2.0 ^b^	1.8 ^bc^	1.7 ^bcd^	1.5 ^cd^	1.4 ^d^	2.0 ^b^	2.4 ^a^
Root length (mm)	67.7 ^a^	64.2 ^ab^	67.5 ^a^	59.4 ^bc^	52.2 ^d^	69.5 ^a^	57.7 ^cd^
Shoot height (mm)	17.5 ^a^	12.2 ^d^	12.0 ^d^	12.4 ^cd^	12.4 ^cd^	14.0 ^bc^	14.4 ^b^
**‘Cerovački Piskavac’**							
Rooting rate (%)	75.6 ^b^	95.6 ^a^	84.5 ^ab^	84.5 ^ab^	84.5 ^ab^	86.7 ^ab^	86.7 ^ab^
No. of roots	2.0 ^cd^	1.8 ^d^	2.4 ^cd^	3.3 ^b^	4.4 ^a^	2.3 ^cd^	2.7 ^bc^
Root length (mm)	64.6 ^a^	55.6 ^bc^	58.1 ^b^	46.8 ^d^	40.9 ^e^	52.0 ^cd^	56.0 ^bc^
Shoot height (mm)	12.9	13.2	13.5	13.8	15.1	13.0	13.5

For each genotype, mean values of each parameter (rooting rate, number of roots, root length, and rooted shoots height) within a single row (labelled with the same small letter in the superscript) were not significantly different according to Duncan’s multiple range test (*p* ≤ 0.05). If the values are not followed with small letters at all, they were not significantly different. +LN—cryo-preserved explants; −LN—non-cryo-preserved explants. All shoots, except those regenerated from dissection controls, were originated from specimens that were pre-cultured for 7 days on a medium containing 0.5 M sucrose in complete darkness at 4 °C; hormonal composition of media used for rooting was: 1 mg L^−1^ IBA and 0.1 mg L^−1^ GA_3_ (‘Belošljiva’, ‘Trnovača’, ‘Dragačevka’, and ‘Cerovački Piskavac’); 1 mg L^−1^ NAA and 0.1 mg L^−1^ GA_3_ (‘Sitnica’, ‘Požegača’, ‘Crnošljiva’, and ‘Moravka’).

## Data Availability

Not applicable.

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
