# Peer review of "Cryopreservation of Indigenous Plums and Monitoring of Multiplication and Rooting Capacity of Shoots Obtained from Cryopreserved Specimens"

_plants, 2023, doi:10.3390/plants12173108_

Round 1

Reviewer 1 Report

The review of “Cryopreservation of indigenous plums and monitoring of multiplication and rooting capacity of shoots obtained from cryopreserved specimens” for Plants MDPI. The topic of manuscript fits within the scope of the journal Plants MDPI.

This paper investigates the possibility to cryopreserved in vitro-grown shoot tips of eight autochthonous Prunus domestica L. from Serbia genotypes using both V cryo-plate and D cryo-plate methods. The relevance of the study is due to the need to preserve the biological diversity of autochthonous plum varieties using modern methods of plant biotechnology (in vitro culture and cryopreservation), which provides an additional guarantee against accidental loss of genetic resources. In the section "Materials and Methods" of the manuscript, the methods and approaches of the conducted research are presented in detail. The results of the study are described in detail and clearly presented in the form of tables and figures (photos in the section "Supplemental material"). A broad discussion of the results obtained with the involvement of a sufficient number of literary sources is given in the section «Discussion» of manuscript.

Thus, this manuscript is an original and completed study. No significant and critical comments were found when reading the manuscript. The manuscript is designed according to the requirements of the journal Plants MDPI.

Reviewer 2 Report

The authors present an interesting study on "Cryopreservation of indigenous plums and monitoring of multiplication and rooting capacity of shoots obtained from cryopreserved specimens'.  It is great to see rooted plants at the end of the study to show the entire process from shoot tip dissection to cryo to rooting.

There are a few minor edits to be made within the manuscript as in terms of tense.  Some examples of these are line 48 was to were, line 50 the word bread is missplet, line 52 has instead of is, line 72 delete etc.  

The introduction can be further improved by giving the readers some examples of diseases which affect Plum on line 50.  Also, the authors should include a justification as why cant plums be seed banked?  i.e recalcitrance. 

The results section should be just the results line 109-117 are not results, this is more M &M. Also the results section contains parts which should be in discussion and removed to make the results section more concise.  

What is the definition of regrowth?? I think a photo reference for the reader would be beneficial. 

English is generally good.  If the authors can just change the tense in sections of the paper. 

Reviewer 3 Report

Dear authors,

I have revised your manuscript “Cryopreservation of indigenous plums and monitoring of multiplication and rooting capacity of shoots obtained from cryopreserved specimens" submitted for publication in Plants. This is an interesting manuscript describing the application of two cryopreservation methods (V cryo-plate and D cryo-plate) for the long-term conservation of Prunus genotypes. Different aspects within each method were accessed and demonstrated to be decisive for the improvement of shoot tip regrowth after cryopreservation. I am suggesting minor modifications that could improve the quality of the document before publication – please see my comments/suggestions below.

I suggest adding the figures in the main document instead of supplementary material

Line 10: vitrification cryo-plate (V cryo-plate) and dehydration cryo-plate (D cryo-plate)

Line 15: plant vitrification solution 2 (PVS2)

Line 16: plant vitrification solution 3 (PVS3)

Line 17: for 2, 2.5, or 3 h

Line 22: at 4°C

Line 298: Add the information regarding the different auxins used at the table footer

• 1 mg l-1 IBA and 0.1 mg l-1 GA3 in: ‘Belošljiva’, ‘Trnovača’, ‘Dragačevka’, and ‘Ce-608 rovački Piskavac’: 609

• 1 mg l-1 NAA and 0.1 mg l-1 GA3 in: ‘Sitnica’, ‘Požegača’, ‘Crnošljiva’, and ‘Moravka’.

Line 320: Consider adding additional references here –

https://doi.org/10.3390/agronomy13010219

https://doi.org/10.1007/s11240-020-01846-x

Line 376: Carrying out the PVS2 treatment at 0°C could reduce the detrimental effects of the VS [23] –practical examples of this characteristic is found in mint shoot tips. Consider adding this practical reference –   https://doi.org/10.1016/j.cryobiol.2005.11.003

Line 400: V cryo-plate? Double check this sentence. Consider reviewing this sentence for something like “D cryo-plate method combines calcium-alginate encapsulation on a cryo-plate with dehydration”

Lines 420-421: There is no link between these sentences.. Please double check it. or should you add V cryo-plate instead of D cryo-plate on line 419?

Lines 429-430: no need to start a new paragraph here

Line 430: you can also add that for some species an increase in regrowth was not found after a cold hardening. For example, in grapes https://doi.org/10.21273/HORTSCI13958-19

Line 440: at 4

Line 489: droplet-vitrification

Line 525: Please use liter in uppercase (30 g L-1). Review this throughout the manuscript.

Lines 526 and 530: Add the medium pH

Line 540: Add the medium pH

Line 540: at 4°C (no need to add the plus symbol)

Lines 542-543: Add the pH of the solutions

Line 543: for 20 min at room temperature

Line 547: Add the pH of the loading solutions

Line 560: in MS? Add the pH of the solution

Line 564: 2, 2.5, or 3 h

Line 568: in MS? Add the pH of the solution.. Why was the sucrose concentration of the unloading solution higher in this method?

Line 587: transferred to

Line 589: in the dark at XX (add temperature) for 7 days

Line 630: at 4

Lines 635-643: Please add a conclusion about this aspect instead of adding what you were assessing.

Add a final sentence describing what the next steps are and how these results contribute to future research or the practical application of this technology.
